# Sonophotocatalysis—Limits and Possibilities for Synergistic Effects

**Dirk Paustian [1,2], Marcus Franke [1,2], Michael Stelter [1,2,3] and Patrick Braeutigam [1,2,3,*]**

[1]  Institute for Technical and Environmental Chemistry, Friedrich Schiller University, 07743 Jena, Germany; dirk.paustian@uni-jena.de (D.P.); marcus.franke@uni-jena.de (M.F.); michael.stelter@uni-jena.de (M.S.)

[2]  Center for Energy and Environmental Chemistry (CEEC Jena), Friedrich Schiller University, 07743 Jena, Germany

[3]  Fraunhofer IKTS, Fraunhofer Institute for Ceramic Technologies and Systems, 07629 Hermsdorf, Germany

[*]  Correspondence: patrick.braeutigam@uni-jena.de; Tel.: +49-364-194-8458

**Abstract:** Advanced oxidation processes are promising techniques for water remediation and degradation of micropollutants in aqueous systems. Since single processes such as sonolysis and photocatalysis exhibit limitations, combined AOP systems can enhance degradation efficiency. The present work addresses the synergistic intensification potential of an ultrasound-assisted photocatalysis (sonophotocatalysis) for bisphenol A degradation with a low-frequency sonotrode ($f = 20$ kHz) in a batch-system. The effects of energy input and suspended photocatalyst dosage ($TiO_2$-nanoparticle, $m = 0$–$0.5$ g/L) were investigated. To understand the synergistic effects, the sonication characteristics were investigated by bubble-field analysis, hydrophone measurements, and chemiluminescence of luminol to identify cavitation areas due to the generation of hydroxyl radicals. Comparing the sonophotocatalysis with sonolysis and photocatalysis (incl. mechanical stirring), synergies up to 295% and degradation rates of up to $1.35$ min$^{-1}$ were achieved. Besides the proof of synergistic intensification, the investigation of energy efficiency for a degradation degree of 80% shows that a process optimization can be realized. Thus, it could be demonstrated that there is an effective limit of energy input depending on the $TiO_2$ dosage.

**Keywords:** sonolysis; photocatalysis; sonophotocatalysis; degradation; bisphenol A; synergy





## 1. Introduction

In the last decade, ultrasound-assisted photocatalysis has received a lot of attention in the degradation of micropollutants in aqueous systems such as endocrine disruptors (EDCs), pharmaceutical active compounds (PhACs), and other organic persistent substances [1–3]. As a part of advanced oxidation processes (AOP), sonophotocatalysis produces high reactive oxygen species (ROS, e.g., •OH) to oxidize organic pollutants with the help of heterogenous catalysts and suitable irradiation in combination with acoustic cavitation [4]. The major aim of combined AOP techniques, for example, sonophotocatalysis, is to enhance the overall efficiency of a single AOP degradation by overcoming its limiting factors and disadvantages [5,6].

In the case of heterogenous photocatalysis with suspended catalytic particles, there must be primarily considered (I) the mass transfer of pollutants to and from a catalytic surface (diffusion limited); (II) agglomeration of suspended particles resulting in decreased catalytic surface area; and (III) homogenization to avoid concentration gradients in the reaction volume.

Several studies have shown that the introduction of low-frequency ultrasound (20–40 kHz) can enhance the photocatalytic degradation process and leads to synergistic effects [3,4,7–9].

Due to cavitation phenomena—known as the formation (nucleation), the growth, and the collapse of water-vapor-filled microbubbles—various chemical [10] and mechanical

effects [11,12] occur, which can be useful for synergistic interactions by overcoming the aforementioned photocatalytic limitations [9]. Chemical effects arise from pyrolytic conditions (~5000 K and ~1000 bar) during bubble collapse [13]. It can be used to degrade organic compounds [14,15] or to initiate homolytic splitting of water molecules in ●OH and ●H for generating additional ROS [9,10]. Mechanical effects depend on the collapse characteristics of the cavitation bubble. An asymmetric collapse leads to microjets, a symmetric collapse leads to shockwaves. Both kinds of collapse conditions accelerate the fluid and cause shear forces in the sonicated media, enhancing the overall mass transfer of pollutants and ensuring steady homogenization and particle deagglomeration [11,12,16]. In general, both the chemical and the mechanical effects increase with increasing energy input.

However, does this mean that the sonophotocatalytic degradation process can be steadily enhanced and maximized by increasing the ultrasonic energy? Otherwise, is there any minimum energy input for reaching any synergistic effects? What synergy can be achieved by acoustic cavitation, apart from noncavitating mixing effects? Additionally, what does the synergy tell us about optimized proceeding conditions?

Furthermore, procedural parameters such as the implemented ultrasound transducer (type and frequency), catalytic particle dosage, or the developed reactor concept must be taken into account.

Moreover, chemical parameters (pH-value, radical scavengers, oxidative additives, sort of pollutant, and catalyst) also influence the sonophotocatalytic degradation process. Detailed reports on these dependencies are given in several publications [7–9,17–24] or reviews [3,4,25,26] and are well-discussed.

However, there is a lack of knowledge concerning the sonophotocatalysis regarding economical optimization potentials versus synergistic intensification considering optimized degradation kinetic constants and energy consumption [27,28].

Therefore, this work investigates the dependencies of ultrasound energy input and catalyst dosage on the sonolytic, photocatalytic, and sonophotocatalytic degradation of the model pollutant bisphenol A in water. Since synergistic effects were shown in literature with tip-sonotrodes [29–36], such a transducer type (20 kHz) was coupled with an UVB-LED system (300 nm) to achieve a maximized overlapping of expected cavitation fields and light irradiation. The degradation experiments were carried out with $TiO_2$ (Degussa P25) in a batch reactor. Correlations were made between degradation processes, kinetic constants, and synergy, with respect to the reactor concept. The sonotrode was characterized by analysis of the bubble field, mapping of oxidizing cavitation areas, and by measuring cavitation intensity. Furthermore, the absorption of the light on various $TiO_2$ dosages were approximated. Finally, an evaluation concerning the relationship between degradation efficiency, synergy, energy consumption, and optimal degradation parameters was done.

## 2. Results and Discussion

### 2.1. Sonolysis

For ultrasound-assisted advanced oxidation processes, various transducer systems can be used. Since the mechanical cavitation effects are desired, sonophotocatalytic degradation procedures are commonly performed with low-frequency ultrasound. In this work, a tip-sonotrode was chosen with a circular vibrating area of 1.3 $cm^2$ and a maximal energy density of 95 $W/cm^2$.

The tip-sonotrode is characterized by an accelerated narrow-defined bubble field directly located in front of the vibrating area, resulting in strong mixing effects in the sonicated media (Figure 1a–c). The intensity depends on the applied amplitude between 38 μm (25%$_{max}$) and 114 μm (75%$_{max}$).

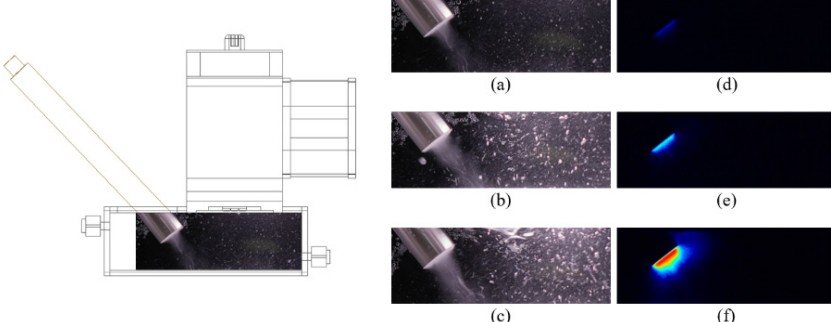

**Figure 1.** Mapping the ultrasound characteristics applied by a tip-sonotrode in water. Bubble-field formation for the amplitudes 38.25, 76.5, and 114.75 μm (**a**–**c**) with generated mixing zones and cavitation zones generating OH radicals visualized by chemiluminescence for the amplitudes 38.25, 76.5, and 114.75 μm (**d**–**f**).

For identical sonolytic conditions, the oxidizing cavitation zones were visualized by the chemiluminescent reaction of luminol due to hydroxyl radical generation (Figure 1d–f). The oxidizing cavitation zones are restricted in front of the vibrating area and correspond partly to the bubble fields. With an intensified bubble field and higher energy input, respectively, an enlargement of the oxidizing cavitation zones can be observed.

This enlargement can be monitored by an increasing cavitation noise level measured with a hydrophone (Figure 2). For the near field around the bubble field there can be derived a linear dependency of the cavitation noise level with the applied amplitude and energy input, respectively. However, if the cavitation noise is considered with further distance to the sonotrode, this linear correlation loses its validity.

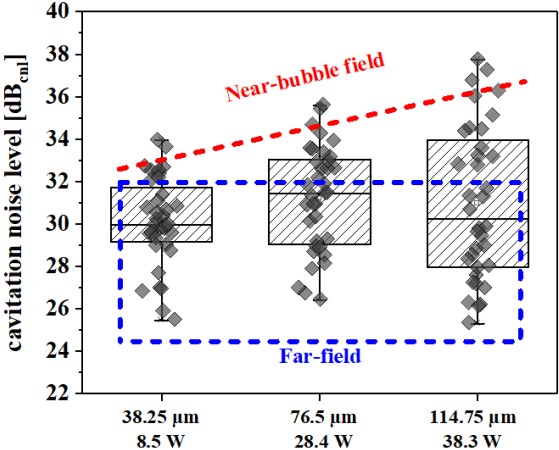

**Figure 2.** Measured cavitation noise level in the sonophotocatalytic reaction zone beneath the UV−LED system.

Thus, it can be concluded that the main cavitation effects occur in the bubble field and in the near area around it. For more distant regions, nonoxidizing cavitation with reduced cavitation intensity can be partly presumed. However, the possibility that the measured cavitation noise is related to acoustic echoes or reflections ("cavitational artifacts") should be kept in mind due to the nonlinear behavior.

The reactive cavitation occurs in the visualized chemiluminescence areas and corresponds to the amount of generated hydroxyl radicals, which can be confirmed by the correlation with kinetic degradation constants (Figure 3). The sonolytic degradation of bisphenol A can be correlated in a linear manner ($R^2$ = 0.995) to the ultrasonic energy input, considering the energy-conversion efficiency (Figure 3a). The kinetic constants of the sonolysis of bisphenol A are calculated in the range of $k_S$ = 0.056–0.091 min$^{-1}$. Furthermore,

it was found that the cavitation noise level also shows a linear correlation to the kinetic constants (Figure 3b).

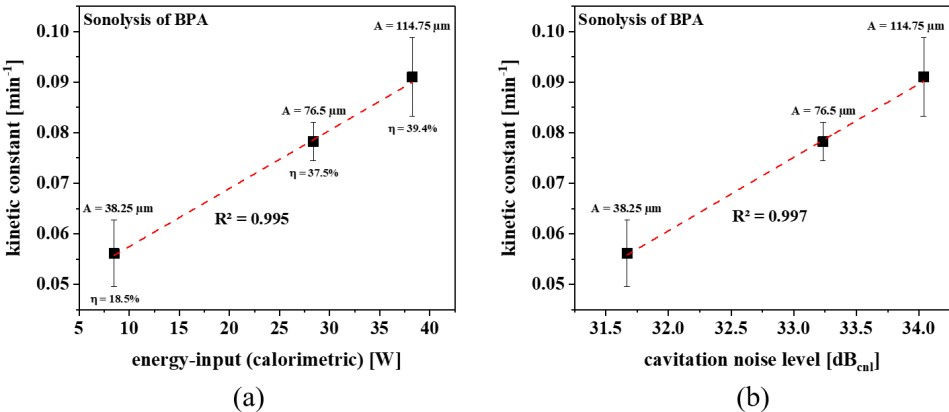

(a)        (b)

**Figure 3.** Correlation between sonolytic rate constants (**a**) and ultrasonic energy input considering the energy−conversion efficiency [η] and (**b**) the cavitation noise level measured in the near field of the oxidizing cavitation area.

## 2.2. Photolysis and Photocatalysis

In this section, the photon-induced degradation of bisphenol A including $TiO_2$ (Degussa P25, $m_{cat.}$ = 0–0.5 g/L) as the photocatalyst and an UVB-LED system (λ = 300 nm) is presented for stationary (without mechanical stirring) and nonstationary (applying mechanical stirring without cavitation effects) systems (Figure 4).

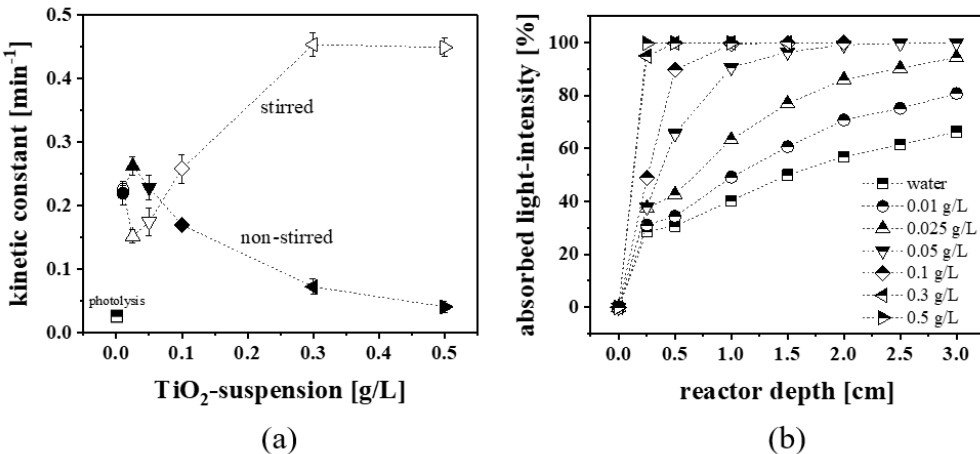

(a)        (b)

**Figure 4.** Photon−induced degradation of bisphenol A and irradiation characteristics. (**a**) photolytic and photocatalytic degradation rates for $TiO_2$ dosages of $m_{cat.}$ = 0–0.5 g/L and λ = 300 nm. (**b**) absorbed light intensity for $TiO_2$ dosages of $m_{cat.}$ = 0–0.5 g/L and λ = 300 nm depending on the reactor depth.

### 2.2.1. Photolysis

The photolysis (without any catalyst loading) shows a degradation rate of $k_P$ = 0.022 $min^{-1}$ (Figure 4a). Due to the absorption behavior of bisphenol A (<285 nm) [37] and the dissociation energy of water for homolytic splitting ($H_2O$ → •OH + •H; dH = 492 kJ/mol ~ <240 nm) [38] to achieve radical oxidizing species, the photolytic degradation of bisphenol A is not effective and can be neglected compared to the photocatalytic degradation procedures with $TiO_2$.

### 2.2.2. Stationary Photocatalysis (Non-Stirred)

After adding $TiO_2$, a strong increase can be noticed with a maximum k-value of $0.262\ \mathrm{min}^{-1}$ for 0.025 g/L $TiO_2$. Further addition of $TiO_2$ results in a decrease in the degradation rate (Figure 4a). The behavior of the photocatalytic degradation can be associated to the UV irradiation and the absorbed light intensity depending on the suspended $TiO_2$ nanoparticles (Figure 4b). Similar to previous studies, it was found that the degradation efficiency strongly depends on the free active catalytic surface and is limited by the turbidity of the suspension [39,40]. With a detailed investigation of the absorbed light intensity of the $TiO_2$ dosages, it could be shown that there are optimal photocatalytic conditions if the whole reactor depth is illuminated and a light intensity of ~10% remains. Although the illumination of the reactor is more efficient for 0.01 g/L $TiO_2$, the available catalytic surface is obviously too low at all. On the other hand, the dosage of 0.05 g/L $TiO_2$ may compensate this limitation, but the estimated illumination of the reactor just takes place in the first centimeter, which is equivalent to 33% of the reactor depth. The remaining reactor can be approximated as a "dead-volume", which is not involved in the degradation process.

Both a lower catalytic surface and/or strong turbidity reflect the main issues in heterogeneous photocatalysis of organic micropollutants, since diffusion is the rate-determining step. To overcome the "dead-volume" issue, the simplest solution is to implement a commercial stirrer to ensure steady homogenization and avoid concentration gradients in the reaction volume

### 2.2.3. Instationary Photocatalysis (Mechanical Stirred)

In this work, a magnetic stirrer was used to investigate a simple method for intensifying photocatalysis without cavitation effects. The mixing speed was chosen so that no vortex occurred. Thus, an enlargement of the illuminated suspension surface was avoided, and the results are comparable to those of the stationary photocatalysis (Figure 4a). It was found that a stirring unit reduces the degradation efficiency for the previously determined optimal reaction conditions at 0.025 g/L and 0.05 g/L $TiO_2$ and enhances the degradation rates for the high suspended reaction solutions of 0.1–0.5 g/L to maximal k-values of $0.450\ \mathrm{min}^{-1}$. It can be concluded that in a stirred system, the limitation of the "dead-volume" issue can be solved and results in higher degradation efficiency for disadvantageous photocatalytic conditions. However, no intensification was observed for photocatalytic conditions, which showed nearly optimal efficiency due to an effective balance between light absorption and catalytic surface area in a stationary system. This can be explained since the UV-LED did not illuminate the whole reactor. Thus, due to stirring, the effective illuminated reaction volume increased and the degradation rates were slightly reduced.

Furthermore, the increasing catalytic surface near the UV irradiation source (higher light intensity) in combination with a steady circulation of the reaction volume has more impact on the degradation efficiency than the illumination of a broader range of the reactor, but with diminished light intensity.

### 2.3. Sonophotocatalysis

In this section, the ultrasonic intensification of the photocatalytic degradation of the model pollutant bisphenol A is investigated and evaluated depending on ultrasonic energy input and $TiO_2$ dosage. Figure 5a,b shows the sonophotocatalytic rate constants compared to those of the stirred photocatalysis for the $TiO_2$ dosages 0.01–0.5 g/L and for the amplitudes 38.25 µm ($A_{US}$ 25%$_{max}$), 76.5 µm ($A_{US}$ 50%$_{max}$), and 114.75 µm ($A_{US}$ 75%$_{max}$).

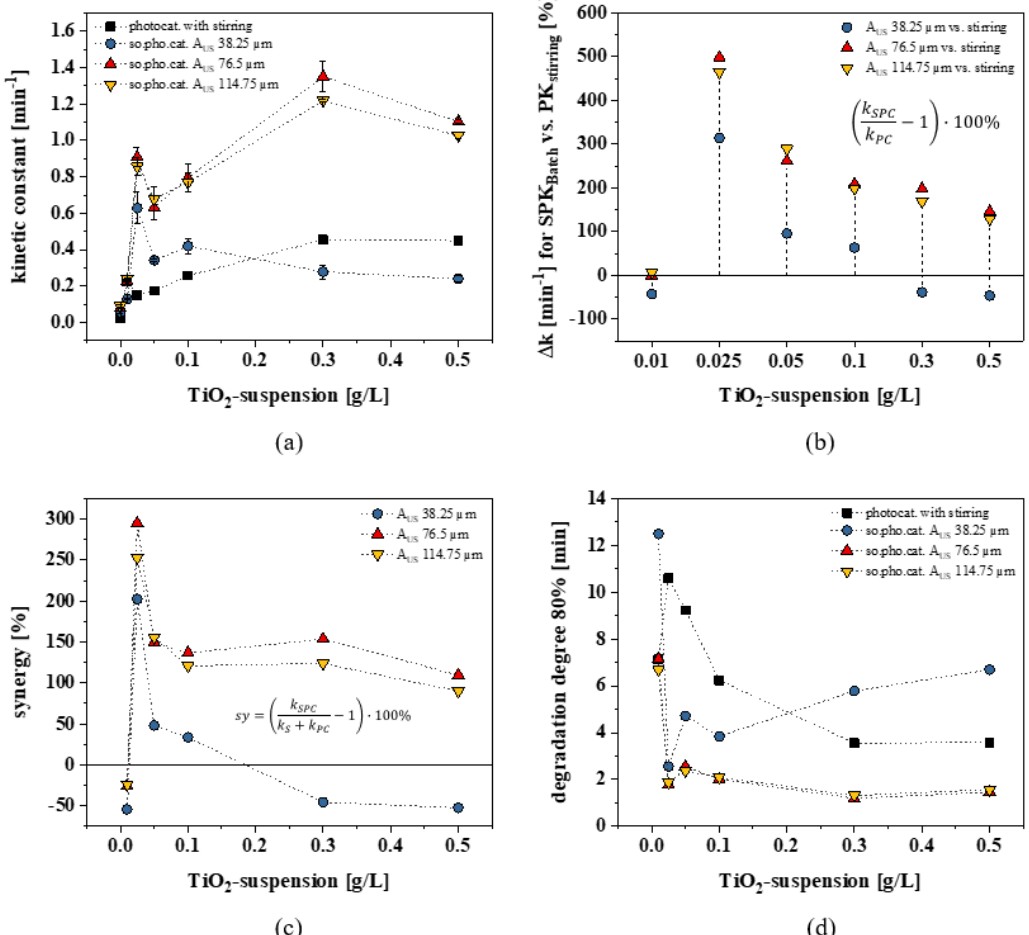

**Figure 5.** Sonophotocatalytic degradation compared to the stirred photocatalysis for the amplitudes 38.25, 76.5, and 114.75 μm and TiO$_2$ dosages 0.01–0.5 g/L. (**a**) Absolute degradation rates; (**b**) difference between photocatalytic and sonophotocatalytic degradation; (**c**) synergistic values; and (**d**) degradation time for 80% degradation degree.

The sonophotocatalytic degradation behavior can be split into two areas depending on the applied TiO$_2$ dosage, consisting of (i) 0.01–0.1 g/L ("low dosage") and (ii) 0.3–0.5 g/L ("high dosage")

### 2.3.1. Low-Dosage Sonophotocatalysis

It can be noticed that the TiO$_2$ dosage of 0.01 g/L is not adequate to achieve further degradation intensifications via sonication. Although at this condition the irradiation is maximized regarding the reaction volume (Figure 4b), the available catalytic surface does not allow any enhancement of photocatalytic degradation via ultrasound. By adding further TiO$_2$ up to a loading of 0.025 g/L, a threshold is exceeded and an interaction with ultrasound leads to strong improvements. Rate constants of k = 0.629 min$^{-1}$ (A$_{US}$ 38.25 μm), 0.908 min$^{-1}$ (A$_{US}$ 76.5 μm), and 0.857 min$^{-1}$ (A$_{US}$ 114.75 μm) were achieved. Comparing the sonophotocatalytic kinetic constants to those of the stirred photocatalysis, the sonication enhances the degradation by 314% (A$_{US}$ 38.25 μm) and 465–499% (A$_{US}$ 76.5 and 114.75 μm) (Figure 5b). Including the sonolytic degradation, synergies of 200% (A$_{US}$ 38.25 μm), 250–300% (A$_{US}$ 76.5 and 114.75 μm) were obtained, and prove the synergistic intensification of the photocatalysis besides mechanical stirring effects (Figure 5c). Furthermore, it can be observed that there is no significant difference between the amplitudes of A$_{US}$ 76.5 μm and 114.75 μm regarding sonophotocatalytic degradation, although the higher energy input leads to more intensive cavitation effects and should enhance the degradation.

Thus, it can be concluded that there is an upper limit for an intensification potential of the sonophotocatalysis (depending on the applied reactor concept) regarding the amplitude and energy input, respectively.

### 2.3.2. High-Dosage Sonophotocatalysis

In the high-dosage suspension range, the overall global maximum of the sonophotocatalysis was found at 0.3 g/L $TiO_2$ with $k_{max,SPC}$ = 1.349 $min^{-1}$ ($A_{US}$ 76.5 μm) and 1.219 $min^{-1}$ ($A_{US}$ 114.75 μm). Compared to the corresponding maximum of the stirred photocatalysis (k = 0.450 $min^{-1}$) with equal $TiO_2$ dosage, the degradation potential was enhanced by ~200% (Figure 5b) with an overall synergy value of ~125–150% (Figure 5c). Further addition of $TiO_2$ leads to a decrease in the degradation, which is attributed to the excessive turbidity and indicates the upper boundary condition for the applied reactor concept similar to the results of the photocatalytic degradation methods. Like the observations in the low-dosage suspension range, the rate constants of $A_{US}$ 76.5 μm and 114.75 μm do not differ significantly. However, for $A_{US}$ 38.5 μm, the sonophotocatalytic degradation decreases below the photocatalytic efficiency. Negative synergy values were obtained (sy = −50%) and indicate the inferior energetical boundary condition for high-suspended reaction solutions. Thus, it is concluded that the low-intensity sonication is not able to interact with the sonophotocatalytic-reaction area since UV irradiation takes place in the first 2.5 mm beneath the LED system. Furthermore, an effective circulation of the high-turbidity suspension cannot be ensured to enhance photocatalytic degradation as it does with the higher energy inputs.

Due to the results of the low- and high-dosage suspended sonophotocatalysis, it can be argued that (i) the enhancing cavitation effects approach a limit, and further increasing the energy does not lead to further enhancements; and/or (ii) the cavitation is immaterial, and the mixing effects majorly achieve synergistic effects and exceed a maximum effective level at $A_{US}$ 76.5 μm.

Comparing Figure 5b,c, a combination of (i) and (ii) can be suggested, since synergistic effects caused by ultrasound were proved besides the raw mechanical stirring effects. Due to the highest difference between stirred photocatalysis and the sonophotocatalysis (0.025 g/L $TiO_2$), it can be presumed that the ultrasound can be effectively used for low-dosage suspensions, even for low ultrasonic energy inputs (Figure 5b). This is underlined by the corresponding maximum synergy values of 200–300% (Figure 5c). However, with increasing $TiO_2$ dosage, it can be approximated that circulation of the reaction media acquires more influence on the degradation. The cavitation effects have a diminished contribution only. This can be especially seen for the $TiO_2$ dosage of 0.3 and 0.5 g/L, where the sonophotocatalysis is less effective than the stirred photocatalysis for $A_{US}$ 38.25 μm (25%$_{max}$), contrary to $A_{US}$ 76.5 μm (50%$_{max}$) and 114.75 μm (75%$_{max}$) (Figure 5a).

According to the degradation behavior, the negative synergies signify that the low-intensity sonication is not appropriate for high-suspended reaction conditions. Nevertheless, with more intensive sonication there can be found positive synergies with 125–150% for 0.3 g/L $TiO_2$. At the same conditions, the overall degradation efficiency reaches its maximum and indicates the optimized reaction conditions, although the synergy value is minor compared to 0.025 g/L $TiO_2$. Thus, a differentiation must be clearly made between degradation potential and synergy for evaluating sonophotocatalytic processes.

### 2.3.3. Energy Assessment

For a target value of 80% degradation, based on the Swiss environmental law for upgrading municipal wastewater-treatment plants [41], the needed reaction times are presented in Figure 5d and were calculated based on the kinetic constants and the pseudo-first-order kinetic by the following equation:

$$t_{80\%} = \left( \frac{\ln 0.2}{-k_{SPC}} \right)$$

Including the energy-consumption calculated by

$$E = \left( \frac{P[kW] \cdot t_{80\%}[h]}{V_R[L]} \right)$$

the efficiency of the sonophotocatalysis is compared to the photocatalysis regarding to the highest degradation rates for the low- and high-suspension range with the TiO$_2$ dosages of 0.025 g/L and 0.3 g/L (Figure 6a,b).

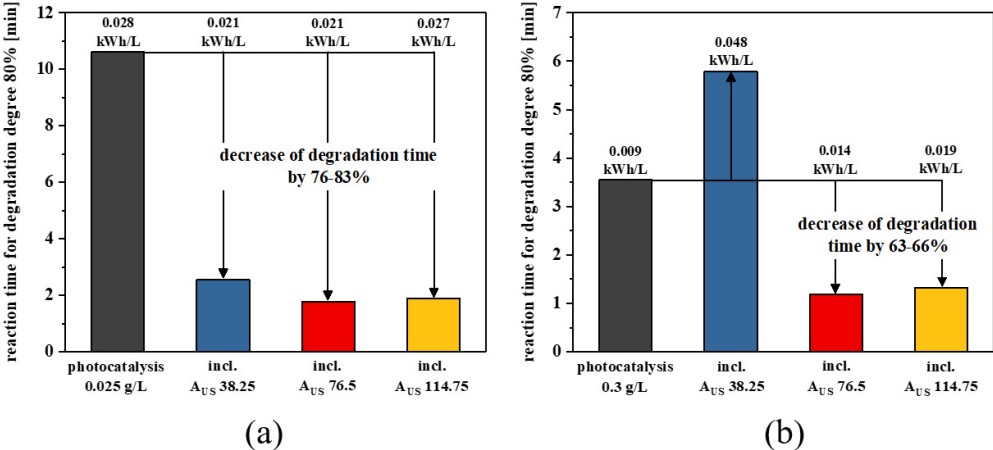

(a)                                                                                                           (b)

**Figure 6.** Reaction time for 80% degradation considering the energy consumption of the irradiation and sonication depending on the amplitude for the TiO$_2$ dosages of (**a**) 0.025 g/L and (**b**) 0.3 g/L.

For the low-dosage suspension, the degradation time was reduced by 76–83% from 10.5 min to 1.9–2.6 min with sonication and led to synergistic effects of 202–295%. It can be noticed that no additional energy is required (Figure 6a). Thus, the degradation time of bisphenol A was effectively reduced without any increasing energy consumption. However, for the higher suspension, additional energy consumption was required to achieve synergistic effects of 154% (A$_{US}$ 76.5 μm) and 124% (A$_{US}$ 114.75 μm) for reducing the treatment time by 66% and 63%, respectively (Figure 6b). Due to the similar degradation results, the optimized sonophotocatalytic process can be found for A$_{US}$ 76.5 μm with extra energy of 55% (0.009 vs. 0.014 kWh/L) compared to the stirred photocatalysis.

### 3. Materials and Methods

*3.1. Materials*

The reagent bisphenol A (>97%) was purchased from Alfa Aesar (Karlsruhe, Germany). Titanium(IV)dioxide nanopowder (P25-Degussa) with primary particle size of 21 nm was purchased from Sigma-Aldrich (Steinheim, Germany). Methanol and acetonitrile were purchased from VWR in HPLC grade. Luminol (97%) and sodium hydroxide were purchased from Sigma-Aldrich (Steinheim, Germany). All chemicals were used without further purification.

*3.2. Setup and Reactor Concept*

The batch reactor contained a tip-sonotrode (Bandelin electronic GmbH & Co. KG, Berlin, Germany, sonopuls GM3200 with generator UW 2200 and SH213G and VS/70T, f$_{US}$ = 20 kHz, A = 1.3 cm$^2$, P$_{A,max}$ = 95 W/cm$^2$) or a magnetic stirrer and a UV-LED-system (Epigap Optronic GmbH, Berlin, Germany, λ = 300 nm, P$_{A,max}$ = 52 mW/cm$^2$) (Figure 7).

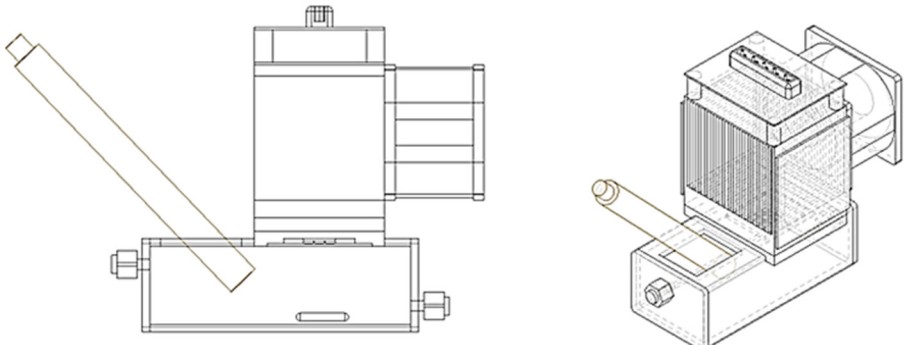

**Figure 7.** Sonophotocatalytic reactor concept (h × b × l = 30 × 42 × 100 mm, V = 134 mL) with tip−sonotrode ($f_{US}$ = 20 kHz, A = 1.3 cm$^2$, $P_{A,max}$ = 95 W/cm$^2$) or magnetic stirrer and a UVB−LED-system ($\lambda$ = 300 nm, $P_{UV}$ = 52 mW/cm$^2$).

The UV system was placed on the top of the reactor. The tip-sonotrode was immerged 1 cm in the sonicated media with an angle of 40 degrees to apply homogenization and cavitation effects in the expected sonophotocatalytic reaction area beneath the UV system.

### 3.3. Mapping of Ultrasound-Induced Bubble Fields

For imaging the bubble fields, a Panasonic Lumix G81 with a Sigma Contemporary F1,4/16 mm was used. Instead of the UV-LED-system an ordinary LED-panel was implemented to illuminate the inner reactor section. The photo was taken through the reactor wall prepared with an acrylic glass plate vertical to the illumination to avoid interferences with the LED panel. To achieve sufficient images, a fast exposure time and a suitable aperture (shutter priority) were adjusted. Afterwards, the images were edited (exposure compensation −2.5) to level the background and work out the bubble fields.

### 3.4. Mapping of Oxidative Species by Chemiluminescence

For imaging the oxidative cavitation zones of the sonotrode, a Panasonic Lumix G81 with a Sigma Contemporary F1,4/16 mm was used. A solution of 170 mg/L luminol and 150 mg/L sodium hydroxide was prepared for the chemiluminescence experiments. The luminol images were taken in complete darkness, from a constant distance and with constant settings (F5,6/128sec/ISO400). With the free image software "ImageJ", the blue-color components were isolated from the original luminol images (RGB-type) and were transformed in lookup tables. In the selected lookup table "Royal", every blue tone (256 color gradations) was replaced by shaded RAL colors resulting in a heat map based on the blue intensity.

### 3.5. Measuring the Cavitation Noise via Hydrophone

Hydrophonic measurements were carried out with the hydrophone Reson TC4043. The software tool from ELMA Schmidbauer GmbH (KaviMeter V5.12.20, Elma GmbH Co. KG, Singen, Germany, 2016) was used to record the cavitation noise level $dB_{cnl}$. Therefore, the reactor section below the light source was divided in 4 layers (each 0.5 mm distance from the reactor bottom) with 9 measured points each layer to create a three-dimensional grid. All the 36 points were arithmetically averaged out of 10 single measurements.

### 3.6. Estimation of Light Absorption by TiO$_2$ Nanoparticles

The light absorption of TiO$_2$ catalyst (0–0.5 g/L) was carried out with a Photometer (NewPort OpticalPowerMeter Model 1916-C with optical detector Model 818-UV-L, Irvine, CA, USA). The UV-LED system was placed on an assembly (4 × 4 cm$^2$) with variable height (0.25–3 cm) containing the suspended nanoparticles. The detector was placed on the bottom of that assembly beneath a quartz glass sheet.

### 3.7. Sono(photo)catalytic Degradation Experiments of Bisphenol A

All degradation experiments were carried out in a batch system with an aqueous bisphenol A solution. For catalytic processes, a specific amount of $TiO_2$ ($m_{TiO2}$ = 0.01–0.5 g/L) was suspended in distilled water and homogenized in an ultrasonic bath (EMAG, Emmi-H60, 40 kHz, 180 W) before adding the bisphenol A to obtain an initial concentration of $c_{BPA}$ = 1 μM. After filling the reactor, the reaction volume was stirred under dark conditions to obtain an absorption equilibrium ($<8\%_{max}$) (Table 1).

**Table 1.** Adsorption behavior of bisphenol A to the reactor and various dosages of $TiO_2$ nanoparticles (Degussa P25) under dark conditions after 1 h with time steps of 15 min.

| m($TiO_2$) [g/L] | Reactor | 0.01 | 0.025 | 0.05 | 0.1 | 0.3 | 0.5 |
|---|---|---|---|---|---|---|---|
| Sorption [%] | $4.51 \pm 0.79$ | $4.90 \pm 0.94$ | $4.52 \pm 1.69$ | $5.69 \pm 0.66$ | $5.34 \pm 2.05$ | $5.67 \pm 1.89$ | $4.01 \pm 2.95$ |

The temperature was held between 20 to 25 °C depending on the energy input of ultrasound. Samples were taken with a needle at up to 5 min reaction time and were immediately centrifuged (Hettich Universal 320 R, 13,500 rpm, $2 \times 5$ min). The quantification of bisphenol A was monitored by HPLC measurements. The kinetic degradation constants were calculated by the pseudo-first-order kinetic as it is postulated for advanced oxidation processes by following equation:

$$\ln\left(\frac{c(t)}{c(0)}\right) = -k \cdot t$$

### 3.8. Quantification of Bisphenol A with HPLC/FD

Quantitative analysis of bisphenol A was carried out with an HPLC-system (JASCO, 2000 series) consisting of the autosampler AS-2055 PLUS, two pumps PU-2080 Plus, the oven 2060 PLUS, the degasser DG-2080/53, and the fluorescence detector FP-2020 PLUS. A reversed-phase C18 column (Dr. Maisch, 250 mm $\times$ 5 mm) was used. An isocratic mobile phase of acetonitrile and water (ratio 65:35) with a flow rate of 1.5 mL/min was set and the oven temperature was held constantly at 40 °C. The emission and excitation wavelengths were 275 nm and 305 nm, respectively. An injection volume of 10 μL with a sample loop of 100 μL was selected.

### 4. Conclusions

In this work, the ultrasonic intensification of a heterogenous photocatalysis with suspended $TiO_2$ nanoparticles via low-frequency ultrasound (20 kHz) was investigated, proved, and evaluated in a batch system. Synergy values up to 200–300%, depending on the energy input of the ultrasound transducer, were obtained in low-suspended (0.025 g/L $TiO_2$) reaction systems. Increasing the catalyst dosage leads to a decrease in the overall synergy if the photocatalytic degradation reaches its maximum. When a $TiO_2$ threshold is exceeded, a certain amount of ultrasonic energy must be applied to continue generating positive synergistic effects. However, it was shown that the synergy does not give a clue for optimized degradation conditions. Despite the decreasing synergy values, the highest degradation rates were obtained with 0.3 g/L $TiO_2$ with a synergy of 100%. Furthermore, it was depicted that the sonophotocatalytic degradation cannot be steadily maximized by applying more and more energy, but there is an effective energy-input limit. Thus, an economical optimization could be derived connected with a process intensification. Aiming for a degradation degree of 80% in low-suspended reaction systems with high synergy values, a reduction in degradation time of 83% was achieved for equal energy consumption compared to the photocatalysis. In high-suspended reaction systems with minor synergy values, an overproportioned reduction in degradation time was achieved regarding the energy consumption.

**Author Contributions:** Conceptualization, D.P. and P.B.; methodology, D.P.; software, D.P.; validation, D.P., P.B. and M.F.; formal analysis, D.P.; investigation, D.P., P.B., M.F. and M.S.; resources, P.B. and M.S.; data curation, D.P.; writing—original draft preparation, D.P. and P.B.; writing—review and editing, D.P., P.B., M.F. and M.S.; visualization, D.P.; supervision, P.B.; project administration, P.B. All authors have read and agreed to the published version of the manuscript.

**Funding:** This research received no external funding.

**Data Availability Statement:** Data are contained within the article.

**Conflicts of Interest:** The authors declare no conflict of interest.

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
