# Peer review of "Sonophotocatalysis—Limits and Possibilities for Synergistic Effects"

_catalysts, doi:10.3390/catal12070754_

Round 1

Reviewer 1 Report

The article reviewed was : Sonophotocatalysis – Limits and possibilities for synergistic effects.

It is well written with very minor grammar and sentence formation errors. The work is not novel but it makes a significant contribution to the field of sonophotochemistry and should be included in this journal. The are 2 items that I would like the authors to look into:

1) Dark experiments were not conducted. TiO2 do have some adsorptive capacity and this could influence the data of the target pollutant in the aqueous solution. I would suggest that the authors presents the adsorption capacity for the TiO2 used in all the experiments in a table.

2) Detail explanations of the data from a chemistry perspectives would be most helpful to the readers and help increased the value of this manuscript as opposed to merely reporting the data

Reviewer 2 Report

In manuscript "Sonophotocatalysis – Limits and possibilities for synergistic effects", the authors have presented synergetic effect of sono- and photocatalysis using commercial TiO2, Degussa P25, as heterogenous catalyst. Authors demonstrated the effects of experimental parameters, such as, stirring, TiO2-dosage, and power of sonotrode, on the overall efficiency of photocatalytic process. An important discovery is the demonstration of the upper limit of ultrasound power required for maximal increase of the efficiency. A large set of data allowed to evaluate the efficiency of the catalytic system in terms of energy consumption, which is fundamentally important for the industrial purification of water from organic pollutants. This manuscript is clearly written and its novelty and importance are beyond doubt. I recommend the publication of this work in Catalyst after minor revisions and some clarifications.

1) The author's affiliation must include the country.

2) page 1, line 17. Add comma before and.

3) page 1, line 32. Since photocatalysts are active not only under UV, I believe that "suitable UV-irradiation" must be replaced with "suitable irradiation".

4) page 3, line 101. Change angel to angle.

5) page 3, line 115. I have some doubts in "luminol pictures" phrase.

6) It looks like (a) and (b) should be deleted from caption to Figure 2.

7) page 5, line 197. I believe that since this sentence is addressed to Figure 4b "the energy input" should be replaced with "cavitation noise level".

8) Figure 6, upper right graph. Caption to X axis in German.

9) page 9, line 318. I also have some doubts in "low suspension ranges" phrase.

10) Please, use uniform style of journal names abbreviations in references section.
